# Early Findings after Implementation of Veno-Arteriovenous ECMO: A Multicenter European Experience

**DOI:** 10.3390/membranes11020081

**Published:** 2021-01-22

**Authors:** Aaron Blandino Ortiz, Mirko Belliato, Lars Mikael Broman, Olivier Lheureux, Maximilian Valentin Malfertheiner, Angela Xini, Federico Pappalardo, Fabio Silvio Taccone

**Affiliations:** 1Department of Intensive Care, Erasme Hospital, Université Libre de Bruxelles, Route de Lennik, 808, B-1070 Brussels, Belgium; ablandinoortiz@gmail.com (A.B.O.); Olivier.Lheureux@erasme.ulb.ac.be (O.L.); angelaxini@hotmail.com (A.X.); 2UOS Advanced Respiratory Intensive Care Unit, UOC Anestesia e Rianimazione 1, Fondazione IRCCS Policlinico San Matteo, 27100 Pavia, Italy; m.belliato@gmail.com; 3ECMO Centre Karolinska, Department of Physiology and Pharmacology, Karolinska Institutet, Karolinska University Hospital, 17177 Stockholm, Sweden; lars.broman@karolinska.se; 4Department of Internal Medicine II, Cardiology and Pneumology, Intensive Care, University Medical Center Regensburg, 93053 Regensburg, Germany; maxmalfertheiner@googlemail.com; 5Department of Cardiothoracic Anesthesia and Intensive Care, Advanced Heart Failure and Mechanical Circulatory Support Program, San Raffaele Hospital, Vita-Salute University, 20121 Milan, Italy; fedepappa.71@gmail.com

**Keywords:** ECMO, veno-artero-venous, differential hypoxia, low cardiac output

## Abstract

Extracorporeal membrane oxygenation (ECMO) is increasingly used to treat cardiopulmonary failure in critically ill patients. Peripheral cannulation may be complicated by a persistent low cardiac output in case of veno-venous cannulation (VV-ECMO) or by differential hypoxia (e.g., lower PaO_2_ in the upper than in the lower body) in case of veno-arterial cannulation (VA-ECMO) and severe impairment of pulmonary function associated with cardiac recovery. The treatment of such complications remains challenging. We report the early effects of the use of veno-arterial-venous (V-AV) ECMO in this setting. Methods: Retrospective analysis including patients from five different European ECMO centers (January 2013 to December 2016) who required V-AV ECMO. We collected demographic data as well as comorbidities and ECMO characteristics, hemodynamics, and arterial blood gas values before and immediately after (i.e., within 2 h) V-AV implementation. Results: A total of 32 patients (age 53 (interquartiles, IQRs: 31–59) years) were identified: 16 were initially supported with VA-ECMO and 16 with VV-ECMO. The median time to V-AV conversion was 2 (1–5) days. After V-AV implantation, heart rate and norepinephrine dose significantly decreased, while PaO_2_ and SaO_2_ significantly increased compared to baseline values. Lactate levels significantly decreased from 3.9 (2.3–7.1) to 2.8 (1.4–4.4) mmol/L (*p* = 0.048). A significant increase in the overall ECMO blood flow (from 4.5 (3.8–5.0) to 4.9 (4.3–5.9) L/min; *p* < 0.01) was observed, with 3.0 (2.5–3.2) L/min for the arterial and 2.8 (2.1–3.6) L/min for the venous return flows. Conclusions: In ECMO patients with differential hypoxia or persistently low cardiac output syndrome, V-AV conversion was associated with improvement in some hemodynamic and respiratory parameters. A significant increase in the overall ECMO blood flow was also observed, with similar flow distributed into the arterial and venous return cannulas.

## 1. Introduction

Extracorporeal membrane oxygenation (ECMO) is increasingly used to treat cardiopulmonary failure in critically ill patients with promising results [1]. Several technological advances, including pump design, more biocompatible cannulas, and larger surface oxygenators, have enabled longer duration of ECMO use and the extension of ECMO indications to a larger group of patients, which have contributed to the development of high-volume centers, new ECMO programs and particular expertise in ECMO management [2]. With the increasing number of treated patients, most patients are percutaneously cannulated [3]; although this is an invasive approach that requires adequate training, the rate of related complications is relatively low and the impact of such complications on mortality remains limited [4].

Peripheral cannulation may be complicated by a persistent low cardiac output in case of veno-venous cannulation (VV-ECMO) or by differential hypoxia (i.e., lower PaO_2_ in the upper than in the lower body) in case of veno-arterial cannulation (VA-ECMO) and severe impairment of pulmonary function associated with cardiac recovery [5]. The treatment of such complications remains challenging, but the hybrid ECMO modes, using a third or fourth cannula, may provide additional support when traditional VV or VA configurations fail to ensure adequate tissue perfusion and oxygenation [6]. In this setting, the veno-arteriovenous configuration (V-AV), where the return outflow (return cannula) is divided in two flows, one toward the aorta to provide circulatory support and the second toward the right atrium to provide respiratory support, has been shown to effectively control these complications related to the initial ECMO configuration [7]; however, no data have been reported on the early effects of V-AV ECMO on patients’ physiology in this setting.

As such, the aim of this study was to report early hemodynamics, gas exchanges, and ECMO characteristics with the use of V-AV in a multicentric case series.

## 2. Materials and Methods

### 2.1. Study Population

This retrospective study was performed in five different European intensive care units (Brussels, Regensburg, Stockholm, Pavia, and Milan), with an ECMO program available according to local protocols. The study protocol was approved by each local ethical committee; in all centers, informed consent was waived because of the retrospective nature of the analysis and because no additional data other than those available in the medical files were collected. All patients treated with V-AV ECMO in the five centers were identified from medical charts and institutional ECMO databases from January 2013 to December 2016.

### 2.2. Data Collection

The demographics and comorbidities (i.e., hypertension, diabetes, respiratory diseases, history of ischemic cardiac disease, pre-existing cardiac or renal failure, liver cirrhosis, solid or hematologic cancer, immunosuppressive therapy, and previous neurological diseases, which could have caused cognitive or other neurovascular disturbance) were collected. The duration of ECMO and intensive care unit (ICU) length of stay were collected. Initial ECMO cannulation (i.e., either VA or VV) and the time required to convert toward V-AV cannulation was recorded; in particular, site of cannulation was also collected. The initial indication for ECMO initiation as well as for ECMO conversion to V-AV configuration were collected. Several data were collected before and within 2 h from V-AV ECMO initiation, including heart rate (HR), mean arterial pressure (MAP), oxygen saturation (i.e., arterial, SaO_2_ or venous, either mixed (SvO_2_) or central (ScvO_2_)), central venous pressure, pulmonary occlusive pressure (PAOP, whenever available), body temperature, arterial lactate, hemoglobin, serum creatinine, daily urine output, the use of renal replacement therapy (RRT), tidal volume, positive end-expiratory pressure (PEEP), pH, PaCO_2_ and PaO_2_, the fraction of inspired oxygen on the ventilator (FiO_2_), the use of neuro-muscular blocking agents (NMBAs), inhaled nitric oxide (iNO), norepinephrine, and dobutamine. The severity of the underlying disease was assessed using the Sequential Organ Failure Assessment (SOFA) score [8]. ECMO characteristics included blood flow, sweep gas flow, and the fraction of oxygen (F_E_O_2_). The presence of distal limb ischemia was also recorded. Mortality was collected at ICU discharge.

### 2.3. Study Outcomes

The primary outcome of this study was to assess changes in hemodynamics and gas exchanges after the initiation of V-AV ECMO. Secondary outcomes included the ECMO configuration (i.e., arterial and venous flow) after V-AV and differences in settings as well as ICU mortality according to the initial ECMO configuration (i.e., VA vs. VV).

### 2.4. Statistical Procedures

Data were tested for normality and are presented as median (interquartile) or mean (±standard deviation), as appropriate. Categorical variables are presented as *n* (%). Categorical variables were compared using Fisher’s exact test, and the Mann–Whitney U-test was used to compare continuous variables. Changes in different variables before and after V-AV initiation were analyzed using the Wilcoxon test. All tests were two-tailed and a *p*-value less than 0.05 was considered statistically significant. Statistical analyses were performed using a SPSS program (IBM SPSS Statistics 24.0 for Windows).

## 3. Results

A total of 32 patients (age 53-IQRs: 31–59-years; 23, 72% male; Table 1) with available data were identified; median time from ICU admission to ECMO was 0 (0–2) days. Sixteen patients were initially supported with VA-ECMO (cardiac arrest, *n* = 6; cardiogenic shock, *n* = 7; septic shock, *n* = 2; myocarditis due to influenza virus, *n* = 1) and 16 with VV-ECMO (acute respiratory distress syndrome (ARDS), *n* = 14; bacterial pneumonia, *n* = 2). The median time to V-AV conversion was 2 (1–5) days, including five patients suffering from ARDS and concomitant cardiogenic shock being converted immediately to this configuration (i.e., <12 h). The reasons for V-AV conversion were differential hypoxia (*n* = 16) in VA-ECMO and biventricular heart failure (*n* = 6), right ventricular failure (*n* = 4), or left ventricular failure (*n* = 6) associated with cardiogenic shock in VV-ECMO.

After V-AV implantation, HR, lactate levels, and norepinephrine dose significantly decreased compared to baseline values (Table 1); moreover, PaO_2_ and SaO_2_ significantly increased while tidal volume and FiO_2_ decreased from baseline. Six patients were on dobutamine before conversion and remained treated after V-AV conversion without significant changes in drug doses (from 17 (range: 13–20) mcg/kg·min to 10 (range: 6–10) mcg/kg·min; *p* = 0.12). The cardiovascular SOFA score remained unchanged (4 (3–4) to 3 (3–4); *p* = 0.20).

A significant increase in the total ECMO blood flow was observed; in particular, 3.0 (2.5–3.2) L/min for the arterial and 2.8 (2.1–3.6) L/min for the venous return flows were used. Other ECMO parameters remained unchanged. For all patients on initial VV configuration, the arterial return cannula was placed into the femoral artery; for the patients with an initial VA configuration, the venous return cannula was placed either in the jugular (*n* = 13) or the femoral vein (*n* = 3).

There were no significant differences in patients on VA- or VV-ECMO as initial configuration for baseline characteristics (Table 2), except for a more frequent use of iNO in VA ECMO patients than others.

In patients on initial VA-ECMO, the implementation of V-AV ECMO was associated with a significant decrease in HR, respiratory rate, tidal volume, and FiO_2_, as well as a significant increase in SaO_2_ and ECMO blood flow (Figure 1, Table 3).

In patients on initial VV-ECMO, the implementation of V-AV ECMO was associated with a significant decrease in HR and respiratory rate, as well as a significant increase in ECMO blood flow (Figure 2, Table 3).

No early specific complication (i.e., bleeding, cannulation failure) was reported. The number of patients with clinical evidence of distal limb ischemia remained similar (5/32, 16% vs. 7/32, 22%; *p* = 0.75). Overall ICU mortality was 19/32 (59%; 10/16, 63% for initial V-A and 9/16, 56% for initial VV configuration).

## 4. Discussion

This multicentric retrospective study showed that the implementation of V-AV ECMO configuration was associated with improved hemodynamics (i.e., reduced HR and norepinephrine requirements, reduced lactate levels) and improved gas exchanges and respiratory mechanics (i.e., increased PaO_2_ and SaO_2_, reduced tidal volume and FiO_2_). The initial ECMO settings assessment showed a similar blood flow in both the arterial and venous return cannulas, with no significant changes in sweep gas flow or oxygen administration via the artificial membrane.

This triple cannulation is not novel but remains a poorly described ECMO configuration, which is usually implemented into an existing VV- or VA -ECMO circuit in specific conditions. Triple cannulation also includes the use of two venous cannulas to drain a higher blood volume and thus increase the amount of oxygenated blood for severe hypoxemia in patients with hyperdynamic status or the overall circulatory support in case of cardiogenic shock and persistent tissue hypoxia or pulmonary hypertension [9]. Despite some potential benefits from such cannulation strategy, scarce data on the hemodynamic and respiratory consequences of V-AV ECMO are available, in particular during the early phase after implementation, where physicians need to stabilize very severe patients. In our series, V-AV ECMO showed to be an effective option to treat ECMO patients with combined cardiopulmonary failure and persistent hypoxemia or tissue hypoperfusion after the initial VA- or VV-ECMO implementation. Other small case series [7,9,10,11,12,13,14,15] have also reported encouraging results with reasonable survival rate, ranging from 25% to 61% (Table 4) for these complex patients requiring hybrid configuration and with a low probability of survival, although the heterogeneity of underlying conditions, patients’ demographics and characteristics, the under-report of complications, and the use of additional therapies, such as long-term mechanical assist devices or organ transplantation, prevent any further analyses by pooling all existing data. Moreover, these reports did not focus on the early effects of V-AV implementation on systemic hemodynamics and gas exchanges, which reflect the immediate effects of ECMO configuration changes on vital functions.

V-AV ECMO should therefore be considered a technically feasible rescue strategy for the treatment of patients suffering from combined respiratory and hemodynamic failure and requiring extracorporeal life support. The configuration mode used for ECMO should be decided upon based on the support required; in some patients, V-AV could also be considered as the initial configuration, although we lack clinical data about the feasibility and effectiveness of this approach. Physicians should not consider V-AV configuration in all patients with a coexistence of cardiac and respiratory failures; optimizing the ventilator settings and reducing pulmonary fluid overload could be sufficient to support the respiratory function in some patients on VA-ECMO with severe hypoxemia. Inotropic agents and the reduction of ventricular afterload may help to improve systemic hemodynamics in patients with VV-ECMO and low cardiac output.

In this study, the indication for V-AV in patients on cardiac support (i.e., VA ECMO) was the presence of differential hypoxia. This phenomenon occurs when the oxygenated blood infused into the femoral artery via the ECMO mixes in the mid-aorta with deoxygenated blood ejected from the native ventricle due to concomitant lung injury [5]. The location of this mixing point depends on the relationship between the ECMO blood flow and the left ventricular ejection (i.e., if severe myocardial dysfunction, the mixing point is closer to the ascending aorta). In our cohort, we did not specifically report diagnostic criteria for differential hypoxia; however, these patients had relatively low PaO_2_ in relationship with high oxygen therapy, both on the ventilator and ECMO. Different interventions, including ventilator adjustments, increasing ECMO flow, central cannulation, or beta-blocking agents, have been proposed to overcome differential hypoxia [16]. We did not specifically report whether some of these therapies had been initiated in these patients before deciding on V-AV conversion; as such, future studies should evaluate the effectiveness and safety of all these potential interventions to further clarify the optimal timing to consider V-AV configuration in this setting. The development of low cardiac output in patients undergoing extracorporeal respiratory support, which was the indication for V-AV conversion in those patients with initial VV-ECMO in this study, has been more rarely reported; important information from estimated cardiac output and/or echocardiography was lacking in our cohort, so it remains difficult to identify the hemodynamic profile or the intensity of medical therapies, which would require the conversion to V-AV ECMO because of persistent tissue hypoperfusion.

As each patient with a V-AV ECMO is required to have an individualized ECMO setting to supply both the respiratory and circulatory function, the analysis of larger registries or cohorts would be helpful to better understand indications, timing of implementation, and management of this ECMO strategy. In the V-AV ECMO configuration, flow through the two return cannulas should be balanced using dedicated flow sensors or adjustable clamps according to the patient’s needs. Routine control of right and left ventricular function by echocardiography, which was missing in our study, would be critical for the fine-tuning of the flow as well as oxygenator and sweep parameters. Moreover, future studies on the effects of such a strategy on patients’ outcomes for those ECMO patients with combined cardiopulmonary failure and/or persistent hypoxemia are required to support its wide use in ECMO centers.

This study presents several limitations. First, because of the rare use of V-AV ECMO, the study cohort was limited; as such, more specific comparisons between subgroups (i.e., VA- vs. VV-ECMO) could not be performed. Moreover, the limited sample size would also limit the robustness of our conclusions and these findings need to be confirmed and replicated in larger cohorts. Second, we only collected ICU mortality, but long-term outcomes are also relevant in ECMO patients. Third, we collected early complications but not others (i.e., infections) that might also occur during the ICU stay. Fourth, we had no information regarding management strategies and decision-making on ECMO initiation, which would be of interest to the readers but beyond the scope of this study. Fifth, we did not report other important issues related to the use of V-AV ECMO, such as cannula size, patients’ monitoring (arterial saturation, tissue oximetry, arterial lines placement, monitoring and management of hemolysis), and weaning; all this information would be relevant, but we specifically focused on the early effects of V-AV implementation and studies in future, larger cohorts should more comprehensively describe all the technical and specific aspects of this ECMO configuration in critically ill patients.

## 5. Conclusions

In this small sample size study, conversion to V-AV ECMO was associated with improvement of some hemodynamic and respiratory parameters. A significant increase in the overall ECMO blood flow was also observed, with similar flow distributed into the arterial and venous return cannulas.

## Figures and Tables

**Figure 1 membranes-11-00081-f001:**
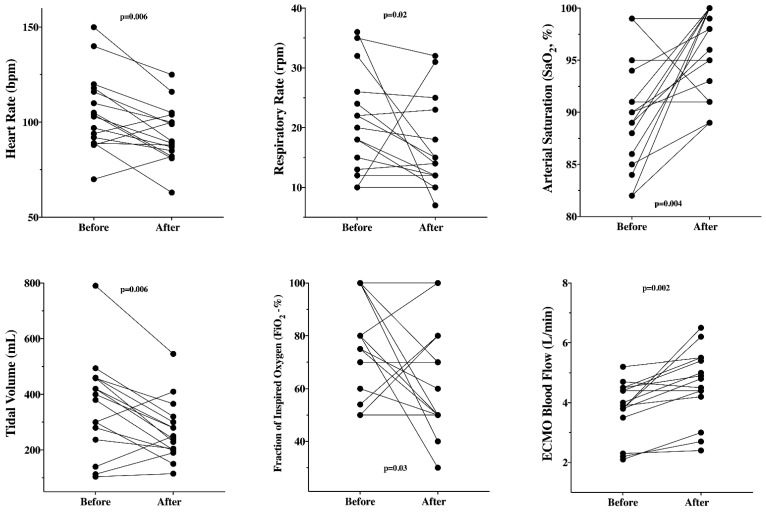
Significant changes in different variables before and after the implementation of V-AV ECMO in patients (*n* = 16) with initial VA-ECMO configuration.

**Figure 2 membranes-11-00081-f002:**
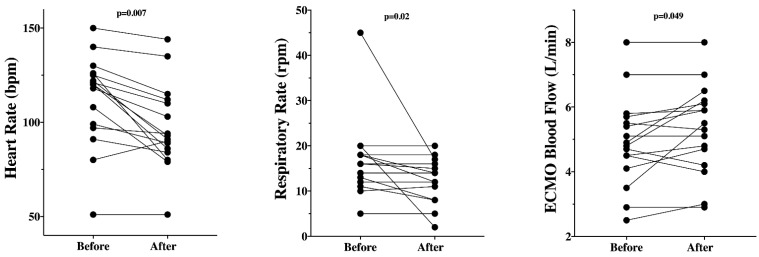
Significant changes in different variables before and after the implementation of V-AV ECMO in those patients (*n* = 16) with initial VV-ECMO configuration.

**Table 1 membranes-11-00081-t001:** Characteristics of the study population. Data are presented as count (percentage) or median (IQRs). BEFORE = initial ECMO support before conversion; AFTER = within 2 h from conversion to veno-arteriovenous (V-AV) mode.

	All Patients (*n* = 32)		
**Age, Years**	**51 (39–59)**		
**Estimated Body Weight, kgs**	85 (77–96)		
**Male Sex, *n* (%)**	23 (72)		
**Days from admission to ECMO**	0 (0–2)		
**Days to V-AV ECMO**	2 (1–5)		
**Total days on ECMO**	9 (5–13)		
**ICU length of stay, days**	14 (9–23)		
**VV/VA ECMO**	16/16		
**Drainage Cannula**			
*femoral*	26		
*jugular*	6		
**Return Cannula**			
*femoral artery*	16		
*jugular vein*	9		
*femoral vein*	6		
*subclavian vein*	1		
**Heart Disease, *n* (%)**	8 (25)		
**Diabetes, *n* (%)**	6 (19)		
**COPD/asthma, *n* (%)**	3 (9)		
**Neurological disease, *n* (%)**	1 (3)		
**Chronic hemodialysis, *n* (%)**	-		
**Liver cirrhosis, *n* (%)**	3 (9)		
**Solid cancer, *n* (%)**	10 (31)		
**Hematological cancer, *n* (%)**	1 (3)		
**Immunosuppressive agents, *n* (%)**	10 (31)		
**CRRT, *n* (%)**	13 (41)		
**NMBAs, *n* (%)**	24 (75)		
**Dobutamine, *n* (%)**	6 (19)		
**iNO, *n* (%)**	8 (25)		
**SOFA score**	14 (12–16)		
**ICU Mortality, *n* (%)**	19 (59)		
	**Before**	**After**	***p*-Value**
**Heart rate, bpm**	109 (94–121)	91 (85–104)	<0.01
**Mean arterial pressure, mmHg**	71 (65–75)	73 (70–75)	0.38
**Central venous pressure, mmHg**	11 (8–13)	11 (9–12)	0.54
**Pulmonary arterial occlusive pressure, mmHg ^£^**	19 (17–21)	20 (14–22)	0.39
**Respiratory rate, rpm**	18 (13–22)	14 (12–16)	<0.01
**Tidal volume, mL**	365 (217–452)	280 (200–400)	0.02
**PEEP, cmH_2_O**	10 (8–14)	10 (8–13)	0.73
**FiO_2_, %**	80 (60–100)	50 (40–80)	<0.01
**SaO_2_, %**	90 (85–96)	95 (92–98)	0.03
**pH**	7.37 (7.28–7.42)	7.38 (7.32–7.44)	0.43
**PaCO_2_, mmHg**	40 (34–43)	37 (35–42)	0.29
**PaO_2_, mmHg**	63 (54–76)	77 (64–92)	0.02
**Temperature, °C**	36.6 (35.7–37.3)	36.6 (36.0–37.1)	0.74
**Lactate, mmol/L**	3.9 (2.3–7.1)	2.8 (1.4–4.4)	0.048
**Hemoglobin, g/dL**	10.1 (9.2–11.2)	10.2 (9.5–11.6)	0.81
**Creatinine, mg/dL**	1.3 (1.0–1.9)	1.3 (1.0–1.9)	0.58
**Norepinephrine dose, mcg/min ^#^**	40 (15–71)	21 (13–37)	<0.01
**ECMO blood flow, L/min**	4.5 (3.8–5.0)	4.9 (4.3–5.9)	<0.01
**ECMO gas flow, L/min**	6 (5–8)	6 (4–8)	0.71
**ECMO F_E_O_2_, %**	100 (100–100)	100 (100–100)	0.99

ECMO = extracorporeal membrane oxygenation; PEEP = positive end-expiratory pressure; ICU = intensive care unit; iNO = inhaled nitric oxide; SaO_2_ = arterial oxygen saturation; SOFA = Sequential Organ Failure Assessment; CRRT = continuous renal replacement therapy; NMBA = neuromuscular blocking agents; COPD = chronic obstructive pulmonary disease; VA = veno-arterial; VV = veno-venous; ^#^
*n* = 27; ^£^
*n* = 16.

**Table 2 membranes-11-00081-t002:** Characteristics of the study population, according to the initial ECMO configuration (VA = veno-arterial; VV = veno-venous). Data are presented as count (percentage) or median [IQRs].

	VA (*n* = 16)	VV (*n* = 16)
**Age, years**	49 (36–58)	51 (41–59)
**Male sex, *n* (%)**	13 (81)	10 (63)
**Days from admission to ECMO**	0 (0–2)	0 (0–4)
**Days to V-AV ECMO**	2 (1–8)	1 (0–3)
**Total days on ECMO**	9 (5–19)	10 (6–12)
**ICU length of stay, days**	14 (8–22)	14 (9–25)
**CRRT, *n* (%)**	7 (44)	6 (38)
**NMBAs, *n* (%)**	14 (88)	10 (63)
**Dobutamine, *n* (%)**	6 (19)	1 (6)
**iNO, *n* (%)**	7 (44)	1 (6) *
**SOFA score**	15 (13–16)	13 (12–15)
**ICU mortality, *n* (%)**	10 (62)	9 (56)
**Heart rate, bpm**	104 (91–117)	120 (99–125)
**Mean arterial pressure, mmHg**	73 (70–76)	67 (60–75)
**Central venous pressure, mmHg**	11 (8–13)	11 (8–12)
**Respiratory rate, rpm**	21 (15–25)	15 (12–18)
**Tidal volume, mL**	390 (269–458)	328 (180–434)
**PEEP, cmH_2_O**	12 (10–14)	9 (7–13)
**FiO_2_, %**	80 (68–100)	90 (50–100)
**SaO_2_, %**	90 (86–93)	93 (79–96)
**pH**	7.39 (7.33–7.44)	7.34 (7.28–7.40)
**PaCO_2_, mmHg**	38 (33–44)	40 (34–43)
**PaO_2_, mmHg**	62 (55–73)	64 (50–91)
**Temperature, °C**	36.6 (36.0–37.2)	36.8 (35.5–37.4)
**Lactate, mmol/L**	3.9 (2.4–5.3)	4.3 (2.3–10.7)
**Hemoglobin, g/dL**	10.2 (9.2–11.1)	10.1 (9.3–11.7)
**Norepinephrine dose, mcg/min ***	43 (23–77) (*n* = 15)	33 (13–71) (*n* = 12)
**ECMO Blood Flow, L/min**	3.9 (3.7–4.5)	4.9 (4.4–5.5)
**ECMO Gas Flow, L/min**	6 (4–8)	7 (6–8)
**ECMO F_E_O_2_, %**	100 (100–100)	100 (100–100)

ECMO = extracorporeal membrane oxygenation; PEEP = positive end-expiratory pressure; ICU = intensive care unit; iNO = inhaled nitric oxide; SaO_2_ = arterial oxygen saturation; SOFA = Sequential Organ Failure Assessment; CRRT = continuous renal replacement therapy; NMBA = neuromuscular blocking agents; * *p* < 0.05 vs. baseline.

**Table 3 membranes-11-00081-t003:** Changes in all measured variables before and after V-AV ECMO implementation, according to the initial ECMO configuration (VA = veno-arterial; VV = veno-venous). Data are presented as median [IQRs].

**VA ECMO**	**Before (*n* = 16)**	**After (*n* = 16)**	***p*-Value**
**Heart Rate, bpm**	104 (91–117)	89 (84–103)	<0.01
**Mean Arterial Pressure, mmHg**	73 (70–76)	72 (68–75)	0.23
**Central Venous Pressure, mmHg**	11 (8–13)	10 (8–12)	0.18
**Respiratory Rate, rpm**	21 (15–25)	15 (12–21)	0.02
**Tidal Volume, mL**	390 (269–458)	245 (200–315)	<0.01
**PEEP, cmH_2_O**	12 (10–14)	11 (8–13)	0.45
**FiO_2_, %**	80 (68–100)	50 (43–78)	0.03
**SaO_2_, %**	90 (86–93)	97 (92–100)	<0.01
**pH**	7.39 (7.33–7.44)	7.39 (7.33–7.47)	0.49
**PaCO_2_, mmHg**	38 (33–44)	37 (34–41)	0.59
**PaO_2_, mmHg**	62 (55–73)	75 (61–101)	<0.01
**Temperature, °C**	36.6 (36.0–37.2)	36.7 (35.6–37.2)	0.55
**Lactate, mmol/L**	3.9 (2.4–5.3)	2.9 (1.2–4.1)	0.28
**Hemoglobin, g/dL**	10.2 (9.2–11.1)	10.1 (9.4–11.5)	0.78
**Norepinephrine Dose, mcg/min**	43 (23–77) (*n* = 15)	25 (9–61) (*n* = 15)	0.16
**ECMO Blood Flow, L/min**	3.9 (3.7–4.5)	4.9 (4.3–5.5)	<0.01
**ECMO Gas Flow, L/min**	6 (4–8)	7 (4–8)	0.21
**ECMO F_E_O_2_, %**	100 (100–100)	100 (100–100)	0.99
**VV ECMO**	**Before (*n* = 16)**	**After (*n* = 16)**	***p*-Value**
**Heart Rate, bpm**	120 (99–125)	92 (85–111)	<0.01
**Mean Arterial Pressure, mmHg**	67 (60–75)	73 (70–80)	0.17
**Central Venous Pressure, mmHg**	11 (8–12)	9 (7–11)	0.07
**Respiratory Rate, rpm**	15 (12–18)	13 (9–20)	0.02
**Tidal Volume, mL**	328 (180–434)	305 (120–450)	0.46
**PEEP, cmH_2_O**	9 (7–13)	9 (8–13)	0.83
**FiO_2_, %**	90 (50–100)	60 (37–95)	0.12
**SaO_2_, %**	93 (79–96)	95 (88–98)	0.13
**pH**	7.34 (7.28–7.40)	7.37 (7.27–7.42)	0.85
**PaCO_2_, mmHg**	40 (34–43)	37 (35–43)	0.28
**PaO_2_, mmHg**	64 (50–91)	77 (66–92)	0.47
**Temperature, °C**	36.8 (35.5–37.4)	37.0 (36.3–37.3)	0.67
**Lactate, mmol/L**	4.3 (2.3–10.7)	2.8 (1.5–8.9)	0.15
**Hemoglobin, g/dL**	10.1 (9.3–11.7)	10.0 (9.2–11.3)	0.87
**Norepinephrine Dose, mcg/min**	33 (13–71) (*n* = 12)	21 (12–30) (*n* = 12)	0.10
**ECMO Blood Flow, L/min**	4.9 (4.4–5.5)	5.4 (4.3–6.2)	0.049
**ECMO Gas Flow, L/min**	7 (6–8)	6 (3–8)	0.67
**ECMO F_E_O_2_, %**	100 (100–100)	100 (100–100)	0.99

ECMO = extracorporeal membrane oxygenation; PEEP = positive end-expiratory pressure; SaO_2_ = arterial oxygen saturation.

**Table 4 membranes-11-00081-t004:** Summary of studies reporting the use of veno-arteriovenous extracorporeal membrane oxygenation on adult patients.

Study (REF)	Type	N	Adults (%)	Data (Years)	Initial ECMO (VA, VV, V-AV)	ECPR	Mortality
**Werner** [11]	R	31	23 (74)	14	2/31 (31%) 8/31 (26%) 11/31 (35%)	9/23 (39%)	14/23 (61%)
**Biscotti** [7]	R	21	21 (100)	2	8/21 (38%) 2/21 (10%) 11/21 (52%)	7/21 (33%)	12/21 (57%)
**Ius** [10]	R	10	10 (100)	3	9/10 (90%) 1/10 (10%) 0	0	5/10 (50%)
**Stöhr** [12]	P	11	11 (100)	3	3/11 (27%) 5/11 (45%) 3/11 (27%)	0	3/11 (27%)
**Vogel** [13]	R	12	12 (100)	3	7/12 (67%) 0 0	5/12 (42%)	3/12 (25%)
**Cakici** [14]	R	12	12 (100)	2	9/12 (75%) 0 0	2/12 (8%)	4/12 (33%)
**Yeo** [15]	R	8	8 (100)	3	8/8 (100%) 0 0	0	4/8 (50%)

P = prospective; R = retrospective; ECPR = extracorporeal membrane oxygenation.

## Data Availability

The database will be available upon request to the authors.

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
