# Peer review of "Early Findings after Implementation of Veno-Arteriovenous ECMO: A Multicenter European Experience"

_membranes, 2021, doi:10.3390/membranes11020081_

Round 1
Reviewer 1 Report
In this well written study the authors investigated the immediate changes that occurred with the insertion of an extra cannula in patients on either V-V ECMO or V-A ECMO. They found an improvement in hemodynamic and respiratory parameters, and an increased flow.
My main problem is that you present your results in a rather general way. This makes it difficult for the reader to assess them. Therefore, please state the specific reasons for the intervention. You touch the subject in the discussion e.g ln 244, and 237-243. This could be in the format of a small table and may help the reader in the interpretation of the results.
Figures 1 and 2 are instructive but limited to significant effects. Given the small sample size you likely throw away information. I suggest to add the effects in Table 2 in the same way as you did in Table 1. In this way the results are visible at a glance for both groups and can easily be compared with the overall results in Table 1.
Please provide in Table 1 the actual P-values for all before/after items.
Table 3 is very difficult to read, even with the references at hand. Moreover, this table does not add anything that is not already in the text. Please omit.
Minor:
You omitted lactate significance in Table 1, but see also above.
Please use spelling and grammar check.
Statistical procedures: Wilcoxon
ln 124 SOFA remained unchanged… Perhaps add: ….in this small sample.
ln 156 difficult to read. One tends to add the flows, please adapt.
Reviewer 2 Report
In this paper the Authors investigated the effects on hemodynamic and respiratory parameters. The title is trustworthy, the aims and methods are clear and conclusions are consistent with results.
Minor comments
In order to enrich the cohort, the enrolled period should be extended (i.g. from January 2013 to December 2019)
Wilconox —> Wilcoxon
Results. As all data were reported in Table 1, there is no need to repeat them in this paragraph. For the same reason Fig1 and Table 1 (Before/After) give similar info. Consider to show the significative parameters’ change in Fig1 and summarize the other the did not (significative) change.
Fig1 and 2. In order to better understand data distribution a plot ,or better, a notched box-and-whiskers are mandatory.
Round 2
Reviewer 1 Report
I am satisfied with the answers on my questions .and remarks.
I have no further comments
Reviewer 2 Report
4. Results. As all data were reported in Table 1, there is no need to repeat them in this paragraph. For the same reason Fig1 and Table 1 (Before/After) give similar info. Consider to show the significative parameters’ change in Fig1 and summarize the other they did not (significative) change.
Authors’ response: We understand this criticism; however, considering the exploratory scope of this study, we have preferred to emphasize most relevant results in the text and Tables, accordingly. We hope the reviewer would understand this choice.
There is not any reason to emphasize the findings in Results section. All (significative and not) findings must be reported avoiding redundances and showing them in the most synthetic and clear way possible. If you think that some issues are important and deserve a special emphasys, use the Discussion section.
5. Fig1 and 2. In order to better understand data distribution a plot ,or better, a notched box-and-whiskers are mandatory.
Authors’ response: Figures have been changed, accordingly.
From the plots, I suppose that the sample size is not sufficent to strongly support the results. You can verify it with a notched box and whiskers plot. Otherwise add this point to limitations.
Author Response
- There is not any reason to emphasize the findings in Results section. All (significative and not) findings must be reported avoiding redundances and showing them in the most synthetic and clear way possible. If you think that some issues are important and deserve a special emphasys, use the Discussion section.
Authors' response: Results have been modified accordingly.
2. From the plots, I suppose that the sample size is not sufficent to strongly support the results. You can verify it with a notched box and whiskers plot. Otherwise add this point to limitations.
Authors response: The authors is right as the sample size is limited. We have emphasized this issue into the Limitation section, accordingly.
Round 3
Reviewer 2 Report
All comments were addressed